# Probing hyperbolic and surface phonon-polaritons in 2D materials using Raman spectroscopy

Alaric Bergeron [1], Clément Gradziel [1], Richard Leonelli [2] &
Sébastien Francoeur [1] ✉

The hyperbolic dispersion relation of phonon-polaritons (PhPols) in anisotropic van der Waals materials provides high-momentum states, directional propagation, subdiffractional confinement, large optical density of states, and enhanced light-matter interactions. In this work, we use Raman spectroscopy in the convenient backscattering configuration to probe PhPol in GaSe, a 2D material presenting two hyperbolic regions separated by a *double* reststrahlen band. By varying the incidence angle, dispersion relations are revealed for samples with thicknesses between 200 and 750 nm. Raman spectra simulations confirm the observation of one surface and two extraordinary guided polaritons and match the evolution of PhPol frequency as a function of vertical confinement. GaSe appears to provide relatively low propagation losses and supports confinement factors matching or exceeding those reported for other 2D materials. Resonant excitation close to the 1s exciton singularly exalts the scattering efficiency of PhPols, providing enhanced scattering signals and means to probe the coupling of PhPols to other solid-state excitations.

Strong coupling between an electromagnetic wave and a solid-state polar excitation yields hybrid quasiparticles known as polaritons[1]. Owing to their mixed light-matter characteristics, polaritons express distinctive properties and enable a unique level of control over light, matter, and their interactions. In this regard, 2D materials have been a prolific material platform for the development of polaritonic devices[2]. Phonon-polaritons (PhPols) results from the coupling of polar transverse optical phonons with light[3]. In strongly anisotropic nanostructures, they exhibit a hyperbolic dispersion relation from which high-momentum states become accessible, providing the means to achieve extreme energy confinement, a large density of states enhancements, and subdiffraction imaging in the mid-infrared region of the spectrum[4,5]. Along with the means to control their propagation[6] and topological state[7], their dispersion can also be dynamically shaped[8]. These are some of the few functionalities that can be used to implement low-loss IR nanophotonic circuits[9].

Due to the large wavevector mismatch between free-space propagating photons and polaritons, special means such as prisms, gratings or nanostructures are required to couple PhPol to far-field instrumentation such as light sources and detectors[10]. In recent years, the exploration of PhPol has extensively relied on scattering-type scanning near-field optical microscopy (s-SNOM)[4–8]. This advanced near-field technique provides compelling images of propagating PhPol as a function of frequency, which reveal the $\omega - k$ dispersion relation. Similar information can be obtained with higher spatial but lower energy resolution using electron energy loss spectroscopy (EELS)[11], a technique implemented within a scanning transmission electron microscope. A large fraction of the recent literature on PhPol has focused on a few prototypical material systems, such as h-BN and $\alpha$-MoO$_3$. If their low-loss, long-lived and tightly confined modes undoubtedly position them favorably, all 2D materials sustaining polar vibrational modes exhibit hyperbolic behavior that could further enable the development of infrared polaritonics. The exploration of

[1]Département de génie physique, Polytechnique Montréal, Montréal, Québec H3C 3A7, Canada. [2]Département de physique & Regroupement Québécois sur les Matériaux de Pointe (RQMP), Université de Montréal, Montréal, Québec H2V 0B3, Canada. ✉e-mail: sebastien.francoeur@polymtl.ca

PhPol has so far relied on rather specialized instruments and has been limited to very few host materials with relatively high-frequency PhPol.

In this work, we demonstrate that Raman spectroscopy is also a powerful technique for studying PhPol in Van der Walls (VdW) materials. Recent works have demonstrated that Raman spectroscopy could probe silent polar phonons and phonon-polaritons through the strong cross-material electron-phonon coupling occurring at the symmetry-breaking interfaces of VdW heterostructures under resonant excitation[12-15]. Here, we probe hyperbolic guided and surface polaritons without the need for excitonic resonances, heterostructures or strong electron-phonon interactions. Indeed, owing to the relaxation of selection rules and the deep sub-wavelength confinement in thin samples, dispersion curves, confinement, and interactions with excitonic resonances can be studied in a backscattering configuration without the need for near-field or other wavevector matching strategies. Hence, Raman spectroscopy is an additional and complementary tool for the development of phonon polaritons.

## Results

### Raman spectroscopy of gallium selenide

For this demonstration $\epsilon$-GaSe is selected due to its strong polar resonances and nonlinear characteristics, nested Reststrahlen bands, double type-II hyperbolic regions (Fig. 1a). The $\epsilon$ polytype has an ABA stacking order, no inversion symmetry, and a $D_{3h}^1$ space group[16,17]. The crystal structure of $\epsilon$-GaSe is shown in the inset of Fig. 1c and is detailed in Supplementary Note 1. This layered mono-chalcogenide has a bulk gap at 2.02 eV[18]. It increases to about 2.43 eV in the bilayer[19], which has a distorted, caldera-shaped valence band[20]. GaSe can be grown using various techniques[20,21]. It is one of the best-known nonlinear crystals for near- to far-infrared operation[22]. These nonlinear properties contribute to the Raman scattering efficiency from long-range polarization waves, making GaSe an ideal prototypical system for studying polaritons.

In the commonly used Raman backscattering configuration, conservation of momentum limits interactions with solid-state excitations with wavevector $k$ about twice that of the incident photon ($k \sim 10^5\,\text{cm}^{-1}$). As this is far beyond the light line $k \sim \omega/c$, polaritons lose their photonic character and become mostly mechanical. Hence, bulk polaritons scattering are typically observed, since their first discovery[3], in a near-forward scattering configuration for which low wavevectors ($k \sim 1 \times 10^3\,\text{cm}^{-1}$) can be reached (see Supplementary Note 2 for more details). We demonstrate next that confined polaritons in 2D materials can be, most conveniently, observed in backscattering configuration for thin ($d \ll \lambda_0$) samples and in some special conditions for thick ($d > \lambda_0$) samples, where $\lambda_0$ is the free-space polariton wavelength.

### Modes identification and dispersion of phonon-polaritons

Backscattering Raman measurements were performed on multiple thin exfoliated GaSe samples (see "Methods") as a function of the sample tilt angle ($\theta$ in Fig. 1b). Strong angular dispersion of several Raman modes were observed in regions corresponding to the *double* Reststrahlen bands ($\epsilon_\perp < 0$, $\epsilon_\parallel < 0$, blue band in Fig. 1a) and the upper type-II hyperbolic band ($\epsilon_\perp < 0$, $\epsilon_\parallel > 0$, yellow band). The data presented in Fig. 1b is from a 650-nm sample and is used to identify these modes. At normal incidence ($\theta = 0°$), the expected Raman-active zone-center modes $A_1'^1$ and $A_1'^4$ (non-polar, 134 and 308 cm$^{-1}$, and $E'$(TO) (polar, 213 cm$^{-1}$) dominate the spectrum. By tilting the sample, resonances with non-zero wavevectors in the sample plane ($k_\parallel$) can be probed. Several important changes can be observed. First, the mode $E''$ appears right below $E'$(TO). Forbidden at $\theta = 0$, this non-polar and therefore purely mechanical mode becomes allowed and observable at incidence angles larger than 30°. More interestingly, two relatively broad features can be readily identified, at all angles, in the region between 230 and 255 cm$^{-1}$. In contrast to all other modes, their frequencies do not match the tabulated frequencies of known GaSe phonons ($A_2''$(TO), $A_2''$(LO) and $E'$(LO)). Furthermore, $A_2''$(TO) and $A_2''$(LO) are Raman-forbidden in all scattering configurations and $E'$(LO) is, like $E''$, forbidden in a normal backscattering configuration and should only appear at a large incidence angle ($\theta > 30°$). The angular dispersion of these two features unambiguously reveals their polaritonic character. As presented in Supplementary Note 3, the transverse extraordinary (Te) phonon connects $E'$(TO) to $A_2''$(TO) and the longitudinal extraordinary (Le) phonon connects $A_2''$(LO) to $E_2'$(LO)[23]. Hence, the first dispersive branch located in the double reststrahlen (blue band) and

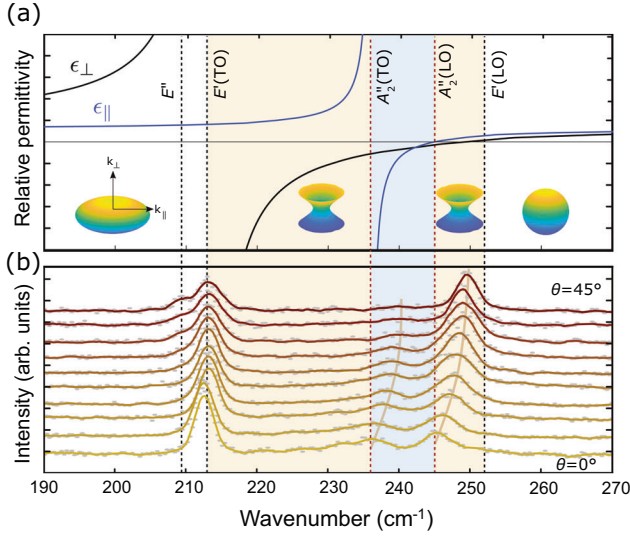

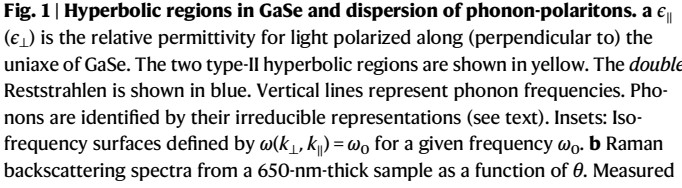

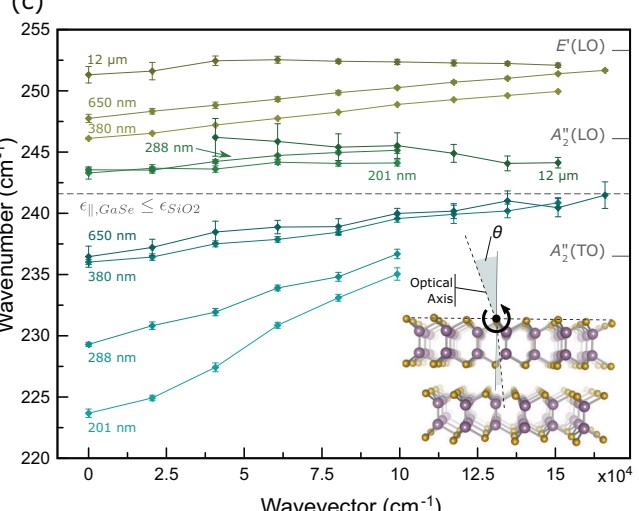

**Fig. 1 | Hyperbolic regions in GaSe and dispersion of phonon-polaritons. a** $\epsilon_\parallel$ ($\epsilon_\perp$) is the relative permittivity for light polarized along (perpendicular to) the uniaxe of GaSe. The two type-II hyperbolic regions are shown in yellow. The *double* Reststrahlen is shown in blue. Vertical lines represent phonon frequencies. Phonons are identified by their irreducible representations (see text). Insets: Isofrequency surfaces defined by $\omega(k_\perp, k_\parallel) = \omega_0$ for a given frequency $\omega_0$. **b** Raman backscattering spectra from a 650-nm-thick sample as a function of $\theta$. Measured data are represented by the gray points, and solid lines are 5-point Savitzky-Golay filtered spectra. The 532-nm excitation is *p*-polarized, and $\theta$ is varied from 0 to 45° in 5° increments. **c** Polariton dispersion as a function of in-plane wavevector ($k_\parallel$) for different sample thicknesses. Blue: lower surface polariton (LSp). Green: upper surface polariton (USp). Yellow: upper extraordinary polariton (UEp). Inset: Crystal structure of $\epsilon$-GaSe and measurement configuration. The tilt angle $\theta$ determines $k_\parallel$.

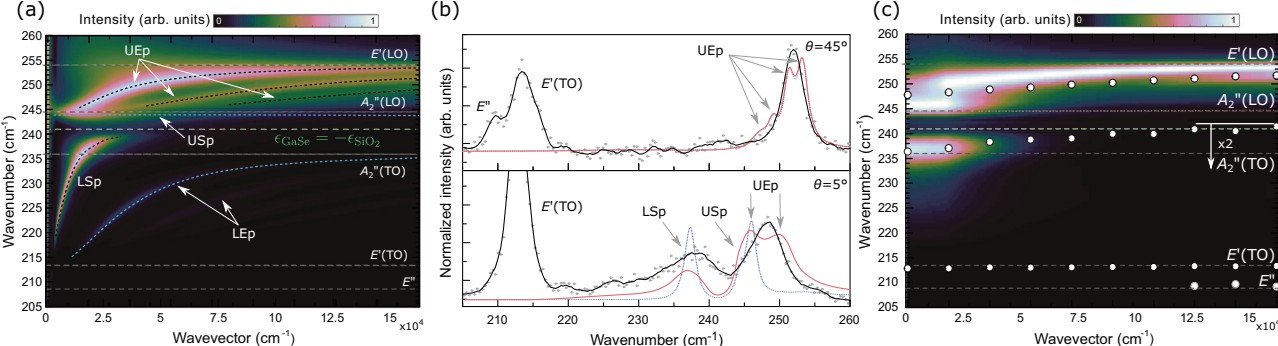

**Fig. 2 | Measured and calculated Raman scattering intensity for a 650-nm GaSe sample on SiO₂/Si. a** Calculated Raman spectra as a function of the propagation wavevector ($k_{\parallel}$). Blue and black lines indicate the polariton modes calculated from the resonances in the imaginary parts of the reflection coefficients.
**b** Experimentally measured (gray data points, black line), calculated (blue line), and corrected for the finite acceptable angle (red line) spectra for $\theta = 5$ and 45°.
**c** Corrected (same as (**a**), but corrected for the finite acceptance angle) and

measured polariton and phonon ($E'$ (TO) and $E''$) dispersions (white circles). For panels (**a**) and (**c**), the intensity scale is logarithmic. For panel (**c**), the intensity has been multiplied by a factor of 2 for frequencies less than 242 cm⁻¹. The frequencies of the zone-center phonons are indicated by the horizontal dashed lines. The green dashed line (**a**, **c**) indicates the lower surface polariton confinement criterion set by the permittivity of SiO₂. UEp and LEp refer to Upper and Lower Extraordinary polaritons. USp and LSp refer to Upper and Lower Surface polaritons.

evolving from 236.4 cm⁻¹ ($\theta = 0°$) to 241.6 cm⁻¹ ($\theta = 45°$) cannot correspond to a phonon. The second branch, evolving from 247.8 to 251.9 cm⁻¹, is in a region where the Le phonon is expected, but the angular shift observed is almost 5 times more significant than that calculated from the lattice anisotropy. Indeed, the phonon dispersion implies a shift of at most 0.9 cm⁻¹[23,24], as illustrated in Supplementary Fig. 2b. The observed shift of 4.1 cm⁻¹ cannot be accounted for using mechanical phonons. Similar Raman spectra as a function of sample tilt were acquired in the manner presented in Fig. 1b on samples of different thicknesses, from thin samples (201 nm) to very thick samples (12 μm) where volume-confined polaritons are not expected. Observed spectral line positions are reported in Fig. 1c as a function of the in-plane propagation wavevector ($k_{\parallel}$). These dispersion curves are organized into three groups (blue, green, yellow) according to their frequency range. Sample thickness has a particularly strong influence on the dispersion of the low and high-frequency groups, a behavior again incompatible with purely mechanical phonons. The data of Fig. 1b, c originate from the backscattering of light from phonon-polaritons (phPols). The strong polaritonic character is further corroborated below through (1) Raman spectra simulations, (2) experimental data demonstrating strong phPol confinement, and (3) experimental resonant Raman data demonstrating enhanced PhPol scattering cross section for excitation close to the GaSe 1*s* exciton.

### Experimental and calculated phonon-polariton spectra
Using the data from the 650-nm sample, we analyze the experimental spectra with the help of calculated Raman polariton spectra. The model, fully detailed in Supplementary Notes 4 to 9, is applicable to any complex anisotropic multilayer structures. The Raman spectra are calculated from[25,26]

$$I(\omega) = C \frac{(n_\omega + 1)}{d} \delta(\Delta q_{\parallel} - k_{\parallel}) \sum_{N = To,Te,Le} H_N^E(\omega,\theta)$$
$$|(\hat{e}_i \cdot \widetilde{R}_N \cdot \hat{e}_s) \int_{-d/2}^{d/2} e^{i\Delta q_\perp z} \langle E_N(z) \rangle dz |^2 , \quad (1)$$

where $N$ refers to the polariton normal coordinates: $To$ for transverse ordinary ($To \perp (\hat{z}, \boldsymbol{k})$), $Te$ for transverse extraordinary ($Te \perp (To, \boldsymbol{k})$), and $Le$ for longitudinal extraordinary ($Le \parallel \boldsymbol{k}$). $\boldsymbol{k}$ is the polariton wavenumber, $C$ is an arbitrary constant, $n_\omega$ is the Bose-Einstein occupation factor, $d$ is the sample thickness, $\hat{e}_{i,s}$ is the polarization of the incident or scattered waves, and $\widetilde{R}_N$ is an effective scattering tensor

taking into account the Raman tensor, the second order susceptibility tensor and the Fröhlich interaction tensor, defined in the polariton normal coordinates as described in ref. 27, but generalized to uniaxial materials. $\langle E_N(z) \rangle$ is the time-averaged root mean square value of the polariton electric field at depth $z$. This field is calculated using a 4×4 transfer matrix formalism fully adapted to multilayered anisotropic media[28]. The Hopfield coefficient $H_N^E(\omega,\theta)$ represents the energy fraction of the polariton stored in the field. The polariton momentum along the surface $k_{\parallel}$ is controlled by the incidence angle ($\theta$) and the change in light momentum is given by $\Delta \boldsymbol{q} = \boldsymbol{q}_i - \boldsymbol{q}_s$. The integral over the scattering depth ($z$) defines the directionality of the scattered light and proves instrumental in understanding the origin of the polariton scattering in both thin and thick samples.

The calculated polariton scattering intensity is shown in Fig. 2a, c as a function of $k_{\parallel}$. Polaritons propagating at oblique angles can exhibit mixed character, and branches are simply identified through their frequency (lower or upper) and character (ordinary, extraordinary or surface). For the calculation of polariton scattering intensity and dispersion, it is found that GaSe polaritons are quite sensitive to their environment, and it is particularly important to consider the whole Si/SiO₂/GaSe/air structure. As illustrated in Supplementary Fig. 3, for example, the polariton field distribution significantly extends into the silicon oxide and substrate, especially at lower incidence angles. Resonances in the imaginary part of the reflectivity coefficients are indicated by the dashed blue lines in Fig. 2a. This more conventional way to calculate surface and guided polariton resonances perfectly agrees with the Raman calculation, but the latter has the advantage to report on the selection rules and the relative intensities relating to scattering.

Figure 2a shows the Raman spectra calculated by summing the $Le$ and $Te$ coordinates. Although the $To$ coordinate reveal the upper and lower ordinary polaritons, they are relatively weak, highly dispersive and very close to the light line (see Supplementary Note 10). These weakly confined polaritons have not been observed, and since the $To$ coordinate does not mix with the other two, it will be ignored. Three types of branches are identified from Fig. 2a: the lower and upper surface (LSp, USp) and (LEp, UEp) extraordinary guided polariton branches. The LEp branch, between the $E'$ (TO) and the $A_2''$ (TO), has a relatively low scattering cross-section. Several UEp branches are found between $A_2''$ (LO) and $E'$ (LO), and these polaritons contribute significantly to the Raman spectra. The two surface branches can be readily identified from their field distribution shown in Supplementary Fig. 3. The USp is located at the GaSe/air interface. Its longitudinal

coordinate dominates, its frequency is only slightly below that of $A_2''$(LO), and its dispersion is weak. The lower surface branch (LSp) is located at the GaSe/SiO$_2$ interface. It shows significant dispersion and scattering cross-section. It starts at the light line, crosses the frequency of $A_2''$(TO) and extrapolates at wavevectors to the frequency determined by the surface mode confinement criterion ($\epsilon_\parallel(\omega_{Sp}) \leq -\epsilon_{SiO_2}(\omega_{Sp})$) indicated by the dashed green line at $\omega_{Sp} = 241\,cm^{-1}$ in Fig. 2a, c. Using calculated frequencies, the three groups reported in Fig. 1c are readily identified as the lower surface (LSp, blue), upper surface (USp, green), and upper extraordinary (UEp, UEp) polaritons.

To directly compare calculated and experimental Raman intensities, the angular selectivity of the Raman measurement system must be considered. The excitation and collection apertures typically used to increase spatial resolution limits the angular resolution required for resolving the dispersion relations. The relatively low numerical aperture of 0.19 of the system corresponds to a collection angle of ±13°. Assuming a Gaussian beam profile, the acceptance angle $\Delta\theta$ at full width at half-maximum is 15°. This limited angular resolution broadens the Raman spectra, especially at small angles where dispersion can be important. This is illustrated in Fig. 2b. At 45°, four UEp contribute to the calculated spectra (dotted blue curve). These modes have a dominant $Le$ coordinate (see Supplementary Fig. 4), and the two high-$k_\parallel$ UEp dominate the spectra with little contribution from the LSp. Experimentally, a weak LSp can be identified and the two dominant UEp appear as a single line (black curve) due in part to a limited signal-to-noise at high measurement angles. Because of the low dispersion at high $\theta$ (high $k_\parallel$), the correction for the finite acceptance angle does not significantly affect the spectra (red curve). At $\theta = 5°$, the calculated Raman spectrum (blue curve) does not quite match the experimental spectra, but the calculated polariton spectra corrected for the acceptance angle (red curve) considerably improve the agreement with the experimental spectrum (black curve). It is important to note that the calculated spectra do not depend on any free adjustable parameters (see Supplementary Note 9), and the agreement with measured spectra is qualitative at low $k_\parallel$ since the linewidths of the LSp and UEp branches are respectively underestimated and overestimated. More work is required to improve the predictive power of Raman intensity calculations at low incidence angles, which is beyond the scope of the present work. In the following, Raman calculations are used as a qualitative tool for the identification of phPols and analysis of their spectra. Hence, it can be said that the unresolved experimental feature at 247 cm$^{-1}$ results predominantly from the upper extraordinary branches (UEP) and, to a lesser extent, the upper surface polariton (USp). The experimental feature at 238 cm$^{-1}$ corresponds to the lower surface polariton (LSp) located at the SiO$_2$ interface. Because of its significant dispersion, the LSp is asymmetric with a low-frequency wing. Finally, on the low-frequency side of the LSp, at about 226 cm$^{-1}$, is a small contribution from the lower extraordinary polariton (LEp). The LEp could not be resolved at any $k_\parallel$, as expected from the low calculated scattering efficiency and finite experimental signal-to-noise ratio.

Figure 2c reports the polariton modes experimentally measured as a function of $k_\parallel$. Data points represent the frequencies identified using a Lorentz-lineshape analysis. The figure also reports the calculated Raman spectra corrected for the finite angular acceptance of the system for comparison. The calculated UEp now appears as a broad feature at low angles, evolving into a single narrower mode extrapolating slightly below $E'$(LO) at high angles. The calculated lower surface polariton LSp starts at 236 and slowly rises toward 241 cm$^{-1}$. Despite not adequately capturing the linewidth of phPol at the lowest $k_\parallel$ (panel (b)), the calculated and measured frequencies are quite consistent, further confirming the identification of guided and surface polaritons. As expected, the latter extrapolates, at high angles, exactly to the frequency determined by the surface confinement criterion discussed above (green dashed line). The calculated intensity of the

lower surface polariton drops off more rapidly than experimentally observed. This may be attributed to the high sensitivity of surface polaritons to the sample surface quality, roughness and contaminant, trapped charges on the SiO$_2$ surface, GaSe oxide, and the overall dielectric environment[29,30].

The integral in Eq. (1) reveals the configuration in which guided polaritons can be observed experimentally. In near-forward, the exponential term is slowly varying since $\Delta q_\perp = q_i \cos\theta_i - q_s \cos\theta_s \sim 0$. Integrated over thick samples $d$, the near-forward selection rule $\Delta q_\perp = k_\perp$ is recovered. In backscattering, $\theta_s \sim \pi + \theta_i$ and $\Delta q_\perp \sim 2q_i$, resulting in rapid oscillations along $z$ and significantly reducing the scattering efficiency from slowly varying polariton fields. Hence, backscattering is precluded from bulk samples, and only lattice phonons ($k \sim 2q_i$) can be measured. However, backscattering can be efficient if the frequency of the polariton field is comparable to $q_i$. For the specific case of GaSe, this occurs for samples thinner than 1 μm (see Supplementary Note 10). This further underscores the fact that the commonly understood association between polaritons and near-forward scattering geometries mostly applies to thick samples. For van der Waals materials, where sub-micron confinement is straightforward, polaritons can be probed in backscattering geometry. Despite this, Raman backscattering from weakly confined polaritons can, in some special conditions, be observed in samples as thick as 70 μm. As demonstrated in Supplementary Note 11, forward scattered Raman signals can be redirected toward the collection optics by a reflection at the back of the sample if the sample is transparent to the scattered light. Hence, care must be applied in the assignment of phonons in 2D materials, as weakly confined polaritons may contribute to the back-scattered Raman spectra for both thin and thick samples.

## Dispersion engineering through confinement

Engineering of the dispersion relations can be achieved by varying the sample thickness, and Raman spectroscopy can be used to investigate both guided and surface polariton frequencies as a function of the sample or waveguide characteristics. Figure 3a presents measured frequencies as a function of sample thickness. The incidence angle is 5°. Although not optimal in terms of spectral resolution, it corresponds to the most convenient and commonly used measurement configuration. In addition, it prevents the observation of the purely mechanical phonons that may become allowed at high angles ($E''$, $E'$(LO)). As in Fig. 2b, c, experimental spectra are dominated by the lower surface (LSp, yellow squares) and upper extraordinary (UEp, blue squares) polaritons. Uncertainties in both frequency and thickness are shown, but for most data points, they are smaller than the symbol size. Figure 3a also presents the calculated scattering efficiency, corrected for the finite acceptance angle, as a function of sample thickness. Each spectrum has been normalized according to its maximum value in order to illustrate the spectral weight rather than the thickness dependence of the intensity. The calculated spectra qualitatively agree with the experimental data. These results demonstrate the relevance of Raman spectroscopy for the investigation of confined polaritons in 2D materials. For some relatively thick samples, two guided UEp can be experimentally resolved. This is the case for the 750-nm-thick sample (see dashed box in Fig. 3a. The corresponding spectrum spectra is shown in the upper panel of 3b (532 nm, 293 K). Indeed, two guided UEp modes are resolved even at 5°. As demonstrated next, these two polaritons are strongly exalted with a Raman excitation close to the excitonic transition.

## Scattering enhancements using resonant excitonic excitations

Raman scattering efficiency is enhanced whenever the incident ($\omega_i$) or scattered ($\omega_s$) photons is resonant with an electronic or excitonic transition ($\omega_X$), as the effective Raman tensor $\widetilde{R}$ presented in Supplementary Note 6 becomes proportional to $(\omega_{i,s} - \omega_X)^{-1}$. For the particular case of the 1s-excitons, a k-dependent Fröhlich contribution

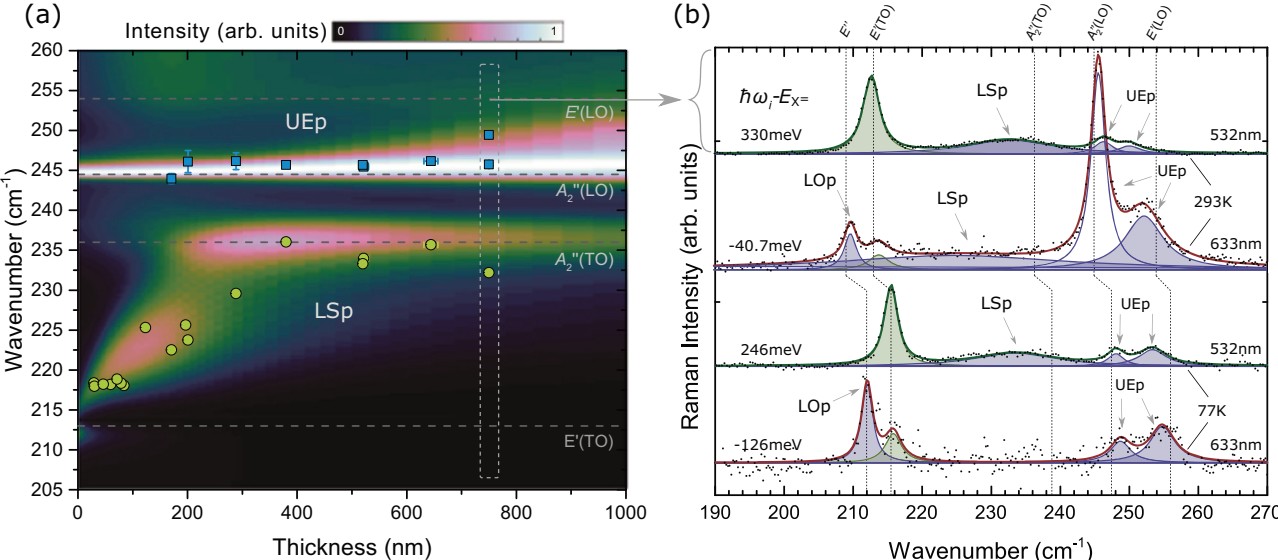

**Fig. 3 | Effects of confinement and quasi-resonant excitation on the phonon-polaritons. a** Upper Extraordinary (UEp, blue squares) and Lower Surface polaritons (LSp, yellow squares) as a function of sample thickness. Around 750 nm (vertical dashed box), two UEp are experimentally resolved, as shown in the upper spectrum in panel (**b**). The color map shows the calculated Raman efficiency corrected for the finite acceptance angle. The incidence angle is 5°. Horizontal lines indicate lattice phonon frequencies. **b** Raman spectra from 750-nm GaSe at 293 and 77 K. Black symbols show the measured data, shaded regions illustrate the Lorentzian contribution from phonons (green) or polariton (purple), and the solid green and red lines represent the fitted spectra for 532 nm and 633 nm laser excitation, respectively. Lattice phonon frequencies are indicated by vertical dashed lines, and the energy difference between the laser excitation ($\hbar\omega_i$) and the 1s exciton ($E_X$) is shown on the left. LOp represents Lower Ordinary polariton.

becomes the dominant intraband scattering mechanism for excitations with a longitudinal component, as demonstrated in Refs. 31,32 and experimentally observed in GaSe for LO phonons[23]. The relatively large exciton binding in bulk GaSe (19.2meV) enables excitonic effects even at room temperature[33,34]. Figure 3b presents Raman spectra from a 750-nm GaSe sample measured as a function of temperature (293 and 77 K) and laser excitation (532 and 633 nm). As demonstrated in ref. 35, temperature-induced phonon shifts result from thermal expansion and three-phonon processes. In the following, however, we analyze the effects of temperature on the Raman intensity, which result from resonant excitation conditions created by temperature *tuning* the 1s-exciton energy ($E_X$) with respect to the laser energy, $\hbar\omega_i$. To that effect, spectra were normalized with respect to that of the totally symmetric and non-polar $A_1'^4$ phonon. The 532 nm excitation exceeds the 1s-exciton by 330 meV (246 meV) at 293 K (77 K). Far away from resonance at both temperatures, the two spectra are almost identical besides the temperature-induced frequency shifts: they both show two upper extraordinary polaritons (UEp) and the surface polariton (Sp). Temperature alone does not qualitatively affect the polariton Raman spectra. In contrast, the 633 nm excitation is only −40.7 meV away from the 1s-exciton at 293 K. This near-resonance excitation induces several striking changes. First, both UEp are significantly exalted, as mentioned above, for excitations with a longitudinal coordinate. At frequencies where the surface polariton is expected, only a very broad continuum is found. Although the lack of a well-defined lineshape prevents a clear identification, the highly dispersive lower extra-ordinary polaritons (LEp) are expected in the spectral region corresponding to LSp and may contribute to the observed signal. Finally, a new polariton appears slightly above the purely mechanical $E''$ phonon, which is forbidden in this normal backscattering geometry ($\theta = 0$) and should only be observed at very high angles ($\theta \geq 35°$). As shown in Fig. 2a for a slightly thinner sample, the lower ordinary polariton (LOp) is expected in this frequency range and has already been identified as such in ref. 36 through forward scattering signals from bulk GaSe. The concomitant effects of low dispersion and resonant enhancements result in a well-defined Raman mode. Cooling the sample to 77 K

moves the 1s-exciton further from the laser excitation, and the intensity from both UEp modes is back to non-resonant levels. In contrast, LOp remains exalted even −126 meV below the 1s-exciton. In fact, this mode has been observed with excitation as far as 410 meV below the exciton[36], indicating that the resonance process involves a much slower decay. These resonant Raman results can provide insights into the coupling between excitonic and polaritonic states[15]. Most significantly, the enhancement of the Raman cross-section, even in quasi-resonance conditions, enables a closer study of polaritonic effects that could not be observed away from resonance due to weaker signals. Under stricter resonance conditions, even greater enhancements are expected.

## Discussion

GaSe is a promising material for polaritonic applications. First, confined phPols are observed in rather thick samples, up to 750 nm, suggesting relatively low propagation losses. Most significantly, high phPol confinement can be achieved. The guided UEp in the 380-nm-thick sample can be observed up to a wavenumber of $k_{||} = 1.5 \times 10^5$ cm$^{-1}$. Far from phonon frequencies and the light line, this polariton is expressing a strongly mixed character, and its confinement factor is estimated between 89 and 110. For the LSp, the confinement factor is between 54 and 79. These factors compare favorably to the best values reported for h-BN and $\alpha$-MoO$_3$[37,38].

We have demonstrated that Raman spectroscopy can be a powerful tool for the study of PhPol and, due to the lack of wavevector matching requirements, conveniently implemented as well. It allows mapping the low energy of phPol (5–150 meV) into the visible where single-photon sensitivity is readily available. It can be applied to a large family of van der Waals crystals[39] with few restrictions on polariton wavevector (>15,000 cm$^{-1}$) and frequency (from 10 cm$^{-1}$ to more than 2000 cm$^{-1}$.) The limited wavevector resolution can be improved by reducing the angular acceptance of the incoming and outgoing light, as demonstrated in Supplementary Note 14, or implementing k-imaging techniques for simultaneously resolving both the frequency and propagation wavevector. Albeit Raman spectroscopy is generally

associated with a weak cross-section, resonant excitation can significantly enhance the scattering efficiency and allows probing excitonic states. Naturally, Raman spectroscopy of polar excitations is restricted to non-centrosymmetric crystals. This includes 3R- and Td-TDMC, for example, as well as odd number of layers h-BN and 2H-TDMC. It also includes crystals for which inversion symmetry is broken[40] by a discontinuity, a perturbation such as an external field, or an interface[13,15]. Almost universally available in 2D material research labs, backscattering Raman spectroscopy is a complementary technique for the study of phPol and may accelerate the development of a wider variety of polaritonic materials and devices.

## Methods

### Sample preparation and protection
Thin samples were first cleaved from Bridgman grown GaSe crystals to expose pristine, optically clear and oxidation-free samples and then transferred onto a PDMS pad. Samples were mechanically exfoliated[41] on a 300-nm thermally grown silicon oxide. The thickness of thin and thick samples was determined from a tapping-mode atomic force microscope or from broadband transmission interferometry using two incidence angles. AFM thickness uncertainties were evaluated from the uncertainties in the position of the top and and bottom surfaces. To avoid sample degradation and photo-oxidation[42], exfoliation and AFM measurements were done in a dry-nitrogen-filled glovebox, and angle-resolved Raman measurements were performed using a vacuum optical cell ($1 \times 10^{-5}$) with low-birefringence windows.

### Backscattering Raman experiments
A typical modular Raman bench was used. Laser excitation was provided by a HeNe (633 nm, ≤1.5 GHz linewidth) or a diode-pumped Nd-YAG (532 nm, 1 MHz linewidth) laser. All spectra were measured using a 532-nm laser, except for those indicated in Fig. 3b. An underfilled objective with a working distance of 9.1 mm provided an effective numerical aperture of 0.19. The sample rotation axis is perpendicular to the incident light beam and positioned at the laser focal point on the sample surface (see inset of Fig. 1c). A goniometer reports the tilt angle with an uncertainty of at most 0.5° over the whole angular range. For polarization control, both the excitation and scattered beam pass through a zero-order $\lambda/2$ waveplate. Parallel or orthogonal detection configuration is selected by adding a polarizer on the path of the scattered beam, followed by a Raman filter and a detection system (spectrometer and CCD camera), providing a raw spectral resolution better than 0.7 cm$^{-1}$. Frequency calibration was based on the omnipresent non-polar modes $A_1^1$ and $A_1^4$. These do not exhibit directional dispersion and are reliable frequency markers. The reported frequency uncertainties are those generated by the nonlinear regression of Lorentz lineshapes. Typically, Raman spectra were acquired using a total exposure time of 15 min using a fluence of the order of 200 μW μm$^{-2}$. All measurements are free from thermal effects.

## Data availability
The spectral data underlying this study are available in the Zenodo database (https://doi.org/10.5281/zenodo.8064143).

## Code availability
The simulation code is available from the corresponding author upon request.

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

## Acknowledgements

This work was supported in part by the Natural Sciences and Engineering Research Council of Canada (SF: NSERC RGPIN-2017-06284 and RGPIN-2023-04675 (S.F.)), Photonique Quantique Québec (SF: *Engineering strongly confined photon-polaritons*), and the Canada Foundation for Innovation (SF: *Platform for the quantum engineering of low-dimensional systems*). A.B. received a Ph.D. scholarship from FRQNT.

## Author contributions

A.B. and C.G. have prepared samples and performed experimental measurements (AFM, optical microscopy, Raman). A.B. has performed all calculations. R.L. and S.F. have supervised the project. S.F. and A.B. wrote the manuscript.

## Competing interests

The authors have no competing interests.
