## [Peer Review File · Nature Communications]

Probing Hyperbolic and Surface Phonon-Polaritons in 2D Materials Using Raman SpectroscopyReviewers' comments:

Reviewer #1 (Remarks to the Author):

****THIS REPORT WAS WRITTEN IN COLLABORATION WITH REFEREE #2**

Review of NCOMMS-22-34152

Bergeron et al. present a very interesting approach of mapping phonon-polaritons with Raman spectroscopy, which is claimed to work because of the broken inversion symmetry of the GaSe crystal. To me this claim of being able to measure infrared-active quasiparticles and their dispersion using the polarizability probe is the central point of their work. I am generally impressed with their work and intrigued by opportunities arising if the claim really holds. If it holds, their work is very important. The key issue overall is the large angular spread of their acquisition system that blurs out the polaritonic features due to momentum blurring, to the extent that a rigorous experimental proof of their claims is in question. Yet, their data and simulations certainly look nice. There are a few points, in the realms of feasibility of their apparatus and approach, that I feel necessary to really convince me:

1) Phonon polariton dispersion: The authors nicely argue why the phonon polaritons should be observable in Raman scattering, and even though my math skills are not sufficient to follow the details, I tend to believe the argument. Clearly, their simulated Raman spectra/dispersion show a behavior that is typical for surface and volume-confined hyperbolic modes, respectively. What I think would strengthen the story, however, is to compare these calculations (and measurements) to "classical" polariton dispersion calculation using the imaginary part of the p-pol reflection coefficient in the infrared. This is the standard approach in the phonon polariton community and can be carried out straight-forwardly for instance using ref. 23. I would expect that the polariton dispersion obtained that way should match up exactly with the calculated dispersion shown in Fig. 2 (curious which of the configurations would fit ...), further convincing me that indeed the authors observe phonon polaritons here. This would additionally make the work more accessible for the phonon polariton community, where this approach is well-established (even though empirically motivated only).

2) Volume-confined hyperbolic modes: The authors try hard to prove the existence of volume-confined modes in the hyperbolic region. However, this is not working well at the moment. For one, higher-order modes are not observed, which is argued being due to the limited resolution (mostly in momentum space due to the finite acceptance angle of the objective). Another way would be to show the layer thickness dependence, which should show a clear trend for volume-confined modes. However, and I am somewhat puzzled by the data in Fig. 3, since this should be measured at non-zero momentum, since only there the modes should show up. So I don't quite understand why Fig. 3 is measured and discussed for normal incidence. I would suggest strongly to calculate and, if possible, also measure the thickness dependence at larger incidence where the higher-order modes and thickness dispersion should be much clearer, indeed supporting the claim. This is clear from Fig. S6, so why measure at 0°?

3) Instrumental perspective: Since the angular spread of the detection seems the major drawback of the presented method, I would hope the authors could sketch a perspective on how to improve this. In the current state of the manuscript and the approach presented, it will be hard to convince other researchers to follow this idea (which I think should be a major prospect when publishing in NatComm). This is a pity since the concept itself is very beautiful.

3a) Possible experimental improvement: I suggest to the authors to implement or suggest in the manuscript the application of a variable aperture in the back focal plane of the microscope objective (or another location at which the conjugate plane is formed). Because this plane is the conjugate or Fourier plane of the image, a variable aperture will allow them to choose the desired angular resolution based on the resulting signal intensity retrieved. In this way, you could then measure the more dispersive modes currently lost in the momentum-integrating scheme currently reported.

4) Resonance Raman: I got lost quite a bit in the discussion of Fig 4. In particular, the Fröhlich

interaction is important in solid-state physics in general, but it is still not clear to me how this connects to polaritons. In particular the sentence "This near-resonance induces several striking changes. First, both UEp are significantly exalted, since the Fröhlich term is particularly important for polaritons with a strong longitudinal coordinate." (Fröhlich is misspelled!) is utterly unclear to me. There are some important recent works extensively discussing hybrid transverse and longitudinal character of polaritons (see for instance <https://doi.org/10.1038/s41467-019-09414-4>). Here, the authors present this in a "everybody knows this" fashion, which I find very confusing. I understand resonance enhancement, but I don't understand the discussion revolving around Fig 4, and how this affects the observation of polaritons by Raman. I strongly suggest to clean up this discussion, and focus on the key message (electronic/excitonic resonance -> signal enhancement). If the authors observe a stronger enhancement of the polariton signals over the phonon signals (which would really be a key observation), they should communicate this more clearly. If not, also ...

5) Higher order modes: Despite of the drawbacks of the instruments, Fig. 2 e and f both show a strong discrepancy between experiment and simulations: two mode orders should be observed in the upper hyperbolic bands at 5° incidence angle even with the large acceptance angle, but only one mode is observed. This is a critical point. In particular, the sentence "However, taking into account the simultaneous contribution of multiple angles in a region of relatively high dispersion yields a calculated polariton Raman scattering spectra (blue curve) matching the experimental one." is simply not true. Overall, I am still hesitant on the major claim. It looks like a convincing story, but the data to me is yet insufficient to really prove it. For the authors approach to really be useful for the wider community, the authors need to fix or at least lay out a clear path of how to fix the issue of limited momentum resolution.

Some minor points:

i) I do not understand why no hyperbolic mode is observed in the lower hyperbolic branch. I see it in the data (no peaks there), but I don't understand it qualitatively/physically. Intuitively, this should be a strong mode.

ii) I am missing a momentum axis in Fig. 2 allowing to relate the experimental parameter (tilt angle) to a proper dispersion (= energy vs. momentum).

iii) I don't quite understand why the GaSe needs to be so thick. In general, volume-confined modes will be harder and harder to be observed, since the propagation across the film gets too long to build up a proper standing wave (for not too small losses).

I understand now from the SI that this is a signal magnitude question. But these two requirements appear contradictory, which would require some discussion. This puts in question the general applicability of the method to other systems, and should be discussed. Typically, thinner layers lead to enhanced confinement, which should at least in part compensate the reduced interaction volume for the Raman process.

iv) I wonder, on a more general level, since the polariton probing with Raman requires broken inversion symmetry, whether it can in some formal way be related to a second-order nonlinear process. I would also be curious if it would ever be feasible to only observe the surface/interface signals from inversion symmetric materials, like often done in sum-frequency generation.

v) There is something missing in the methods section: "[] followed by a Raman filter and a detection system (spectrometer and CCD camera) providing a spectral resolution better than."

Reviewer #2 (Remarks to the Author):

**THIS REPORT WAS WRITTEN IN COLLABORATION WITH REFEREE #1

Review of NCOMMS-22-34152

Bergeron et al. present a very interesting approach of mapping phonon-polaritons with Raman spectroscopy, which is claimed to work because of the broken inversion symmetry of the GaSe crystal. To me this claim of being able to measure infrared-active quasiparticles and their dispersion using the polarizability probe is the central point of their work. I am generally impressed with their work and intrigued by opportunities arising if the claim really holds. If it holds, their work is very important. The key issue overall is the large angular spread of their acquisition system that burrs out the polaritonic features due to momentum blurring, to the extent that a rigorous experimental proof of their claims is in question. Yet, their data and simulations certainly look nice. There are a few points, in the realms of feasibility of their apparatus and approach, that I feel necessary to really convince me:

1) Phonon polariton dispersion: The authors nicely argue why the phonon polaritons should be observable in Raman scattering, and even though my math skills are not sufficient to follow the details, I tend to believe the argument. Clearly, their simulated Raman spectra/dispersion show a behavior that is typical for surface and volume-confined hyperbolic modes, respectively. What I think would strengthen the story, however, is to compare these calculations (and measurements) to "classical" polariton dispersion calculation using the imaginary part of the p-pol reflection coefficient in the infrared. This is the standard approach in the phonon polariton community and can be carried out straight-forwardly for instance using ref. 23. I would expect that the polariton dispersion obtained that way should match up exactly with the calculated dispersion shown in Fig. 2 (curious which of the configurations would fit ...), further convincing me that indeed the authors observe phonon polaritons here. This would additionally make the work more accessible for the phonon polariton community, where this approach is well-established (even though empirically motivated only).

2) Volume-confined hyperbolic modes: The authors try hard to prove the existence of volume-confined modes in the hyperbolic region. However, this is not working well at the moment. For one, higher-order modes are not observed, which is argued being due to the limited resolution (mostly in momentum space due to the finite acceptance angle of the objective). Another way would be to show the layer thickness dependence, which should show a clear trend for volume-confined modes. However, and I am somewhat puzzled by the data in Fig. 3, since this should be measured at non-zero momentum, since only there the modes should show up. So I don't quite understand why Fig. 3 is measured and discussed for normal incidence. I would suggest strongly to calculate and, if possible, also measure the thickness dependence at larger incidence where the higher-order modes and thickness dispersion should be much clearer, indeed supporting the claim. This is clear from Fig. S6, so why measure at 0° ?

3) Instrumental perspective: Since the angular spread of the detection seems the major drawback of the presented method, I would hope the authors could sketch a perspective on how to improve this. In the current state of the manuscript and the approach presented, it will be hard to convince other researchers to follow this idea (which I think should be a major prospect when publishing in NatComm). This is a pity since the concept itself is very beautiful.

3a) Possible experimental improvement: I suggest to the authors to implement or suggest in the manuscript the application of a variable aperture in the back focal plane of the microscope objective (or another location at which the conjugate plane is formed). Because this plane is the conjugate or Fourier plane of the image, a variable aperture will allow them to choose the desired angular resolution based on the resulting signal intensity retrieved. In this way, you could then measure the more dispersive modes currently lost in the momentum-integrating scheme currently reported.

4) Resonance Raman. I got lost quite a bit in the discussion of Fig 4. In particular, the Fröhlich interaction is important in solid-state physics in general, but it is still not clear to me how this connects to polaritons. In particular the sentence "This near-resonance induces several striking changes. First, both UEp are significantly exalted, since the Fröhlich term is particularly important for polaritons with a strong longitudinal coordinate." (Fröhlich is misspelled!) is utterly unclear to me. There are some important recent works extensively discussing hybrid transverse and longitudinal character of polaritons (see for instance <https://doi.org/10.1038/s41467-019-09414-4>). Here, the authors present this in a "everybody knows this" fashion, which I find very confusing. I understand resonance enhancement, but I don't understand the discussion revolving around Fig 4, and how this affects the observation of polaritons by Raman. I strongly suggest to clean up this discussion, and focus on the key message (electronic/excitonic resonance -> signal enhancement). If the authors observe a stronger enhancement of the polariton signals over the phonon signals (which would really be a key observation), they should communicate this more clearly. If not, also

...

5) Higher order modes. Despite of the drawbacks of the instruments, Fig. 2 e and f both show a strong discrepancy between experiment and simulations: two mode orders should be observed in the upper hyperbolic bands at 5° incidence angle even with the large acceptance angle, but only one mode is observed. This is a critical point. In particular, the sentence "However, taking into account the simultaneous contribution of multiple angles in a region of relatively high dispersion yields a calculated polariton Raman scattering spectra (blue curve) matching the experimental one." is simply not true.

Overall, I am still hesitant on the major claim. It looks like a convincing story, but the data to me is yet insufficient to really prove it. For the authors approach to really be useful for the wider community, the authors need to fix or at least lay out a clear path of how to fix the issue of limited momentum resolution.

Some minor points:

i) I do not understand why no hyperbolic mode is observed in the lower hyperbolic branch. I see it in the data (no peaks there), but I don't understand it qualitatively/physically. Intuitively, this should be a strong mode.

2) I am missing a momentum axis in Fig. 2 allowing to relate the experimental parameter (tilt angle) to a proper dispersion (= energy vs. momentum).

3) I don't quite understand why the GaSe needs to be so thick. In general, volume-confined modes will be harder and harder to be observed, since the propagation across the film gets too long to build up a proper standing wave (for not too small losses).

I understand now from the SI that this is a signal magnitude question. But these two requirements appear contradictory, which would require some discussion. This puts in question the general applicability of the method to other systems, and should be discussed. Typically, thinner layers lead to enhanced confinement, which should at least in part compensate the reduced interaction volume for the Raman process.

4) I wonder, on a more general level, since the polariton probing with Raman requires broken inversion symmetry, whether it can in some formal way be related to a second-order nonlinear process. I would also be curious if it would ever be feasible to only observe the surface/interface signals from inversion symmetric materials, like often done in sum-frequency generation.

5) There is something missing in the methods section: "[] followed by a Raman filter and a detection system (spectrometer and CCD camera) providing a spectral resolution better than."

Reviewer #3 (Remarks to the Author):

This paper sets out measurements of Raman scattering from 'thin' exfoliated layers of GaSe in a backscattering geometry as a function of the angle of incidence/collection. The resulting spectra are interpreted in terms of surface phonon and hyperbolic polaritons present in the material. The experiments appear to have been done well, are presented in sufficient detail to be reproduced and the interpretation is entirely believable.

The authors claim the significance of this paper is that it demonstrates the utility of Raman scattering for the study of phonon polaritons and technology based upon these materials. This has already been established by work on Raman scattering from hyperbolic phonon polaritons in hBN and surface phonon polaritons in sapphire via their effect on excitonic states in transition metal dichalcogenide layers (Jin, C., Kim, J., Suh, J. et al. Interlayer electron-phonon coupling in WSe₂/hBN heterostructures. *Nature Phys* 13, 127–131 (2017). <https://doi.org/10.1038/nphys3928>) and papers which cite this paper.)

Another conclusion of the paper is that the thin nature of the samples allows polaritons to be observed in backscattering not just forward scattering. The breakdown of conservation of forward momentum due to thin samples has been known for a very long time, e.g. it in allowing the backscattering geometry for four wave mixing in GaAs quantum wells.

One failing of the paper is that it doesn't sufficiently address the existing literature in the field. This includes the work on Raman scattering from polaritons in exfoliated van der Waals materials. In addition the paper doesn't review the literature on GaSe Raman scattering which includes experiments on 'thin' layers, e.g. Selective Raman modes and strong photoluminescence of gallium selenide flakes on sp² carbon

Journal of Vacuum Science & Technology B 32, 04E106 (2014);
<https://doi.org/10.1116/1.4881995>. Including a review of this work in the paper would enable the reader to have a better sense of the novelty within the results.

In conclusion the paper presents some nice experiments which are well presented but these results are not sufficiently novel for publication in Nature Communications. The existence of phonon polaritons in GaSe is already established; the fact that Raman scattering can be measured from these modes is established; the fact that the dispersion of such modes can be measured by angle dependent Raman scattering is established. I therefore advise the editor not to accept this article for publication and the authors to revise the manuscript to include all the relevant related literature and publish this work in a reputable archive journal.

Reviewer #4 (Remarks to the Author):

In this manuscript, the authors report probing phonon polariton in GaSe thick flakes and want to demonstrate that routine Raman spectroscopy is sufficient for acquiring information for phonon polaritons in certain materials. This manuscript is a well-prepared Raman study on GaSe phonon polariton, including experimental measurement and theoretical calculations. Especially in the supporting information part, the authors provide rich details about the calculation, which is a good reference for those dealing with Raman spectral calculations. However, the experimental evidence in the current manuscript does not fully support the conclusion. The authors present the measured Raman spectra of GaSe as a function of impinging angle in the beginning, which shows the features in the Raman spectra changes with k indeed. However, the rest of the paper, or the essentials of the discussions, are based on pure calculations. The only direct comparison of measurement and calculation in Fig.2e does not show a good agreement, although the authors argued in the main text for the possible causes. This is a good paper showing an interesting case for capturing a few polariton features in two-dimensional materials using a conventional spectroscopic tool, Raman spectroscopy. Nevertheless, the connection between the measurement and calculation is not sound enough to be extrapolated to study phonon polariton properties offered by SNOM extensively. This manuscript is more suitable for a specialized journal. Apart from the general comments, there are more comments and suggestions below for authors to consider.

1. The supporting information pointed out, "For typical phonon frequencies ($\omega \approx 200\text{--}1000\text{ cm}^{-1}$, polaritonic effects are observed for wavevectors values of the order of $k \sim 1 \times 10^3\text{ cm}^{-1}$." As known, the wavevector for the phonon polariton of a 220 nm thick MoO₃ is $k_{\text{polariton}} \sim 1 \times 10^5\text{ cm}^{-1}$, two orders larger than the authors' description. (Nature Communications 11, 2646 (2020)). The authors are encouraged to provide dispersions of the phonon polaritons by calculating the imaginary part of the Fresnel reflection coefficient of the Air/GaSe/SiO₂/Si multilayer system (Nature Nanotech 12, 207–211 (2017)). If the $k_{\text{polariton}}$ is close to or not much smaller than the wavevector of the light, the propagation directions of q_s and q_i in Fig S1c will deviate greatly, and there will be a significant difference between the reflection angles (θ_{os}) and the incident angles (θ_{oi}). When obtaining the phonon's momentum, authors can plot the θ_{os} at different θ_{oi} in Fig S1c.

2. A detailed description of the experimental configuration is missing, including how to adjust a good focus at large oblique angles and how the 15-degree impinging angle is estimated. When the incident angle θ in Fig 1a is 45°, one long-distance objective alone may not collect sufficient Raman scattering signal from the reflection light.

3. The double peak feature for UEp1 and UEp2 is unclear in the experimental data in Fig 1d and Fig 2e, while the calculation predicted a clear double peak feature. In Fig 2f, the authors pointed out, "data points represent the dominant frequencies identified using a Lorentz-line shape analysis of the experimental spectra." In contrast, the calculated spectra did not show Lorentz-lineshape (245 cm^{-1} to 255 cm^{-1}) with the sample tilting angle from 0° to 10°. The author may consider taking the second derivative of the spectrum to locate the peak positions of UEp1 and UEp2.

4. Although Raman spectroscopy shows advantages in detecting phonon polaritons from the far-field, the precision of the dispersion curve depicted by Raman spectroscopy is far from the level SNOM can offer at the moment. In conclusion, the authors should discuss possible improvement routes for future technologies based on the above shortcoming of Raman spectroscopy.

5. In Fig4, the authors should briefly discuss the physical origin of temperature-induced Raman shifts and cite relevant new literature, such as Sci Rep 6, 32236 (2016). Since several textbooks elaborated well on the observation of probing polaritons in GaAs at low temperatures, what is new in Fig.4 should be stressed.
6. In Fig 2f, the green dashed line is missing but mentioned in the caption.
7. Quote from the SI, "phonons can straightforwardly be defined by their A, E, LO and LO characters." LO duplication.
8. Quote from the sentence, " The calculated polariton scattering intensity is shown in 2(a-c,d,e) as a function of sample tilt angle" should be 2(a-c,d,f)
9. Several typos, such as SiO2 printed as Si02.
10. On page 14, the first sentence is incomplete.

Reviewer #1 (Remarks to the Author):

**THIS REPORT WAS WRITTEN IN COLLABORATION WITH REFEREE #2
Review of NCOMMS-22-34152

***COMMENT**

Bergeron et al. present a very interesting approach of mapping phonon-polaritons with Raman spectroscopy, which is claimed to work because of the broken inversion symmetry of the GaSe crystal. To me this claim of being able to measure infrared-active quasiparticles and their dispersion using the polarizability probe is the central point of their work. I am generally impressed with their work and intrigued by opportunities arising if the claim really holds. If it holds, their work is very important.

RESPONSE

We thank reviewers (#1 and #2) for having carefully read our manuscript and taken the time to share with us their comments and suggestions. We would like to apologize for taking several months for providing this response. The first author is a former Ph.D. student that graduated more than one year ago. Despite his strong commitment and desire to see the core of his PhD findings published, it has not been an easy task to collaborate on revising and improving the manuscript with a regular full-time job in a start-up and a young family.

***COMMENT**

The key issue overall is the large angular spread of their acquisition system that blurs out the polaritonic features due to momentum blurring, to the extent that a rigorous experimental proof of their claims is in question. Yet, their data and simulations certainly look nice. There are a few points, in the realms of feasibility of their apparatus and approach, that I feel necessary to really convince me:

Phonon polariton dispersion: The authors nicely argue why the phonon polaritons should be observable in Raman scattering, and even though my math skills are not sufficient to follow the details, I tend to believe the argument. Clearly, their simulated Raman spectra/dispersion show a behavior that is typical for surface and volume-confined hyperbolic modes, respectively. What I think would strengthen the story, however, is to compare these calculations (and measurements) to “classical” polariton dispersion calculation using the imaginary part of the p-pol reflection coefficient in the infrared. This is the standard approach in the phonon polariton community and can be carried out straight-forwardly for instance using ref. 23. I would expect that the polariton dispersion obtained that way should match up exactly with the calculated dispersion shown in Fig. 2 (curious which of the configurations would fit ...), further convincing me that indeed the authors observe phonon polaritons here. This would additionally make the work more accessible for the phonon polariton community, where this approach is well-established (even though empirically motivated only).

RESPONSE

As suggested by Reviewer 1 and 2, we have calculated the polariton dispersion curves from the reflectivity of the whole structure. The results are shown in Fig. 2(a) superimposed onto the calculated Raman spectra. The results perfectly match. The following text was added to the manuscript:

“Resonances in the imaginary part of the reflectivity coefficients are indicated by the dashed blue lines in Panel (a). This more conventional way to calculate surface and guided polariton resonances perfectly agrees with the Raman calculation, but the latter has the advantage to report the selection rules and the relative intensities relating to scattering.”

*COMMENT

Volume-confined hyperbolic modes: The authors try hard to prove the existence of volume-confined modes in the hyperbolic region. However, this is not working well at the moment. For one, higher-order modes are not observed, which is argued being due to the limited resolution (mostly in momentum space due to the finite acceptance angle of the objective). Another way would be to show the layer thickness dependence, which should show a clear trend for volume-confined modes.

RESPONSE

We have added the dispersion curve measured on five samples in Fig. 1(b). Although not all surface and guided modes are observed from all samples, surface and guided polaritons are clearly identifiable. In addition, this experimental data demonstrate that the dispersion relation is strongly influenced by the sample thickness. The following text was added.

“Similar Raman spectra as a function of sample tilt were acquired in the manner presented in Fig. 1a) on samples of different thicknesses, from 201 nm to 12 microns. Observed spectral line positions are reported in Fig. 1(b) as a function of the in-plane propagation wavevector (k_{\parallel}). These dispersion curves are organized in three groups (blue, green, yellow) according to their frequency range. Sample thickness has a particularly strong influence on the dispersion of the low and high energy groups, a behavior again incompatible with purely mechanical phonons. The data of Fig. 1(a-b) originate from the backscattering of light from phonon-polaritons (phPols). The strong polaritonic character is further corroborated below through 1) Raman spectra simulations, 2) experimental data demonstrating strong phPol confinement, and 3) experimental resonant Raman data demonstrating enhanced PhPol scattering cross section for excitation close to the GaSe $1s$ exciton.”

Using this data, we have calculated confinement factors and added the following text.

“GaSe is a promising material for polaritonic applications. First, confined phPols are observed in rather thick samples, up to 12 μ m, suggesting relatively low propagation losses. Most significantly, high phPol confinement can be

achieved. The guided UEp in the 380-nm thick sample can be observed up to a wavenumber of $k_{\parallel} = 1.5e5 \text{ \per\centi\meter}$. Far from phonon frequencies and the light line, this polariton is expressing a strongly mixed character and its confinement factor is estimated between 89 and 110. For the LSp, the confinement factor is between 54 and 79. These factors compare favorably to the best values reported for h-BN and $\alpha\text{-MoO}_3$ [37,38]”

Higher-order guided modes have been so far seldom observed, especially in thin samples, either due to the instrumental broadening or sample quality. That being said, we present in Fig 3(a) and 3(b) data of from a relatively thick sample where two extraordinary guided polaritons could unambiguously be resolved (see dotted box in Fig.3(a)). The four experimental spectra shown in Fig. 3(b) present the behavior of these two modes at two different temperatures and two excitation wavelength. The text was modified to bring out this aspect more clearly.

This demonstrated that at least two modes can be observed. Future work will include enhanced sample preparation methods, increased angular resolution, and improved instrumental sensitivity. This may allow the observing higher order modes more consistently.

***COMMENT**

However, and I am somewhat puzzled by the data in Fig. 3, since this should be measured at non-zero momentum, since only there the modes should show up. So I don't quite understand why Fig. 3 is measured and discussed for normal incidence. I would suggest strongly to calculated and, if possible, also measure the thickness dependence at larger incidence where the higher-order modes and thickness dispersion should be much clearer, indeed supporting the claim. This is clear from Fig. S6, so why measure at 0° ?

RESPONSE

Indeed, broadening is minimized at higher sample tilts. This aspect is clearly demonstrated in Fig. 2(a) (new data). In Fig. 3(a), the measurement angle is 5 degrees for all the data points. The following text was added to clarify our choice,

“Although not optimal in terms of spectral resolution, it corresponds to the most convenient and commonly used measurement configuration. In addition, it prevents the observation of the purely mechanical phonons that may become allowed at high angles ($E^{\prime\prime}$, $E^{\prime}(LO)$)”

As our goal is to demonstrate the Raman measurements of phonon-polaritons, it was in our opinion strategic to avoid configurations that may be contaminated by signals from mechanical phonons. Naturally, further studies will take advantage of the low broadening at high k.

***COMMENT**

Instrumental perspective: Since the angular spread of the detection seems the major drawback of the presented method, I would hope the authors could sketch a perspective on how to improve this. In the current state of the manuscript and the approach presented, it will be hard to convince other researchers to follow this idea (which I think should be a major prospect when publishing in NatComm). This is a pity since the concept itself is very beautiful.

RESPONSE

This research is relatively novel and little efforts have yet been devoted to this aspect by us or other scientists. That being said, the experimental work and related Raman calculations have allowed us to develop the understanding required to explore means to improve the resolution and relevance of the technique. We show in section S14 and Fig. S8 that a simple aperture (as suggested by the reviewer) can improve the resolution of the measurement. We are currently in the processes of implementing a k-imaging setup that would allow measuring the scattered intensity as a function of both frequency and wavevector. The following text was added to the conclusion,

“The limited wavevector resolution can be improved by reducing the angular acceptance of the incoming and outgoing light, as demonstrated in S14 for the LSp, or implementing k-imaging techniques for resolving both the frequency and the propagation wavevector.”

***COMMENT**

Possible experimental improvement: I suggest to the authors to implement or suggest in the manuscript the application of a variable aperture in the back focal plane of the microscope objective (or another location at which the conjugate plane is formed). Because this plane is the conjugate or Fourier plane of the image, a variable aperture will allow them to choose the desired angular resolution based on the resulting signal intensity retrieved. In this way, you could then measure the more dispersive modes currently lost in the momentum-integrating scheme currently reported.

RESPONSE

Please, see response presented above.

***COMMENT**

4) Resonance Raman: I got lost quite a bit in the discussion of Fig 4. In particular, the Fröhlich interaction is important in solid-state physics in general, but it is still not clear to me how this connects to polaritons. In particular the sentence “This near-resonance induces several striking changes. First, both UEp are significantly exalted, since the Fröhlich term is particularly important for polaritons with a strong longitudinal coordinate.” (Fröhlich is misspelled!) is utterly unclear to me. There are some important recent works extensively discussing hybrid transverse and longitudinal character of polaritons (see for instance <https://doi.org/10.1038/s41467-019-09414-4>). Here, the authors present this in a “everybody knows this” fashion, which I find very confusing. I understand resonance

enhancements, but I don't understand the discussion revolving around Fig 4, and how this affects the observation of polaritons by Raman. I strongly suggest to clean up this discussion, and focus on the key message (electronic/excitonic resonance -> signal enhancement). If the authors observe a stronger enhancement of the polariton signals over the phonon signals (which would really be a key observation), they should communicate this more clearly. If not, also ...

RESPONSE

The electron-lattice interaction Hamiltonian is composed of two terms: the deformation potential and the electro-optic (or Fröhlich) terms. The second term relates to the modulation of the materials polarizability by the electric field of the polariton. These two terms are used to define the effective Raman tensor (see Eq. S14). In resonance, both of these mechanisms are enhanced. In addition to these two effects, Martin [31-32] has demonstrated that an additional electro-optic (Fröhlich) contribution must be considered at the excitonic resonance. This k -dependent contribution, equals to zero for $k=0$, has been shown to become a singularly important scattering mechanism for longitudinal excitations (LO phonons or polaritons with a longitudinal coordinate) for $k>0$.

To clarify this aspect, the following sentence was added to the text:

“For the particular case of the 1s-exciton, a k -dependent Fröhlich contribution becomes the dominant intraband scattering mechanism for excitations with a longitudinal component, as demonstrated in Refs. [31-32] and experimentally observed in GaSe for LO phonons [23]. ”

The curious reader will be able to examine the complete derivation of this additional resonant term and the effect it can have on the resonant intensity. Please note that we have not yet been able to calculate the polariton Raman spectra under resonance, as this k -dependent contribution is much more complicated to implement than the other terms.

*COMMENT

Higher order modes: Despite of the drawbacks of the instruments, Fig. 2 e and f both show a strong discrepancy between experiment and simulations: two mode orders should be observed in the upper hyperbolic bands at 5° incidence angle even with the large acceptance angle, but only one mode is observed. This is a critical point. In particular, the sentence “However, taking into account the simultaneous contribution of multiple angles in a region of relatively high dispersion yields a calculated polariton Raman scattering spectra (blue curve) matching the experimental one.” is simply not true.

RESPONSE

Modifications to the text have been made to emphasize the fact that the Raman simulation is mainly used as a qualitative tool for the interpretation of the experimental data.

The discussion on the nature of the discrepancies in Fig. 2(e-f) (previous version) is now better described. In reference to the two panels of Fig 2(c), the following text has been added.

“At 45°, four UEp contribute to the calculated spectra (dotted blue curve). These modes have a dominant L_e coordinate (see Fig. S4) and the two high- k_{\parallel} UEp dominate the spectra with little contribution from the LSp. Experimentally, a weak LSp can be identified and the two dominant UEp appear as a single line (black curve), due in part to a limited signal-to-noise at high measurement angles. Because of the low dispersion at high θ (high k_{\parallel}), the correction for the finite acceptance angle does not significantly affect the spectra (red curve).} At $\theta=5^\circ$, the calculated Raman spectrum (blue curve) does not match the experimental spectra, but the calculated polariton spectra corrected for the acceptance angle (red curve) considerably improve the agreement with the experimental spectrum (black curve). It is important to note that the calculated spectra do not depend on any free adjustable parameters (see Section S9) and the agreement with measured spectra is only qualitative at low k_{\parallel} , since the linewidths of the LSp and UEp branches are respectively underestimated and overestimated. More work is required to improve the predictive power of Raman intensity calculations at low incidence angles, which is beyond the scope of the present work. In the following, Raman calculations are used as a qualitative tool for the identification of phPols and analysis of their spectra.

***COMMENT**

Overall, I am still hesitant on the major claim. It looks like a convincing story, but the data to me is yet insufficient to really prove it. For the authors approach to really be useful for the wider community, the authors need to fix or at least lay out a clear path of how to fix the issue of limited momentum resolution.

RESPONSE

We hope that the additional information and experimental data added to the manuscript will convince the reviewers of the relevance of the technique. The path towards a higher resolution includes techniques such as “k-imaging”.

***COMMENT**

Some minor points:

i) I do not understand why no hyperbolic mode is observed in the lower hyperbolic branch. I see it in the data (no peaks there), but I don't understand it qualitatively/physically. Intuitively, this should be a strong mode.

RESPONSE

Compared to the upper Reststrahlen, polariton fields were much lower in the lower Reststrahlen band. This strongly affects the Raman scattering efficiency of all three polariton coordinates. Unfortunately, we are not able to provide a more intuitive picture at this point.

*COMMENT

ii) I am missing a momentum axis in Fig. 2 allowing to relate the experimental parameter (tilt angle) to a proper dispersion (= energy vs. momentum).

RESPONSE

All dispersion curves are now reported in terms of the propagation wavevector.

*COMMENT

iii) I don't quite understand why the GaSe needs to be so thick. In general, volume-confined modes will be harder and harder to be observed, since the propagation across the film gets too long to build up a proper standing wave (for not too small losses). I understand now from the SI that this is a signal magnitude question. But these two requirements appear contradictory, which would require some discussion. This puts in question the general applicability of the method to other systems, and should be discussed. Typically, thinner layers lead to enhanced confinement, which should at least in part compensate the reduced interaction volume for the Raman process.

RESPONSE

For this study, it has proven easier to work with relatively thick samples, both in terms of sample preparation, sample quality and uniformity, and signal-to-noise ratio. That being said, the confinement data presented in Fig 3(a) covers the range from 30 nm to 750 nm. The fact that we measure phonon-polaritons in relatively thick samples is, as the comment made by the reviewer suggests, a very positive aspect : it indicates that losses in GaSe are relatively low, thereby allowing confinement in relatively thick samples.

To discuss the effect of thickness, the following text was added to section S7 of the Supp. Info.

“As a function of sample thickness, the Raman intensity is governed by the integral of Eq. S19 and the probed volume. At 532 nm, the integral is maximized for a thickness of about 150 nm. Away for this thickness, the integral slowly decreases. The probed volume increases linearly with thickness, but saturates at the penetration depth of the excitation laser (5.6\micrometer).”

*COMMENT

iv) I wonder, on a more general level, since the polariton probing with Raman requires broken inversion symmetry, whether it can in some formal way be related to a second-order nonlinear process. I would also be curious if it would ever be feasible to only

observe the surface/interface signals from inversion symmetric materials, like often done in sum-frequency generation.

RESPONSE

Yes, the electro-optic Raman tensor elements found in Eq. S14 are directly connected to the second harmonic generation tensor. A sentence to that effect can be found in S6. Yes, Raman signals can be generated from surfaces. This aspect has been added in the conclusion.

***v) There is something missing in the methods section: “[] followed by a Raman filter and a detection system (spectrometer and CCD camera) providing a spectral resolution better than.”**

Thanks for reporting this. This has been corrected.

Reviewer #2 (Remarks to the Author):

**THIS REPORT WAS WRITTEN IN COLLABORATION WITH REFEREE #1

See previous answers

Reviewer #3 (Remarks to the Author):

***COMMENT**

This paper sets out measurements of Raman scattering from 'thin' exfoliated layers of GaSe in a backscattering geometry as a function of the angle of incidence/collection. The resulting spectra are interpreted in terms of surface phonon and hyperbolic polaritons present in the material. The experiments appear to have been done well, are presented in sufficient detail to be reproduced and the interpretation is entirely believable.

RESPONSE

We thank Reviewer #3 for having read our manuscript and taken the time to prepare this report. We would like to apologize for taking several months for providing this response. The first author is a former Ph.D. student that graduated more than one year ago. Despite his strong commitment and desire to see the core of his PhD findings published, it has not been an easy task to collaborate on revising and improving the manuscript with a regular full-time job in a start-up and a young family.

COMMENT

The authors claim the significance of this paper is that it demonstrates the utility of Raman scattering for the study of phonon polaritons and technology based upon these materials. This has already been established by work on Raman scattering from hyperbolic phonon polaritons in hBN and surface phonon polaritons in sapphire via their effect on excitonic states in transition metal dichalcogenide layers (Jin, C., Kim, J., Suh, J. et al. Interlayer electron–phonon coupling in WSe₂/hBN heterostructures. *Nature Phys* 13, 127–131 (2017). <https://doi.org/10.1038/nphys3928>) and papers which cite this paper.)

RESPONSE

We were well aware of the work of Jin et al. [R1]. This work reports fascinating results on the electron-phonon coupling (EPC) in atomically thin 2D heterostructures: a WSe₂ monolayer sandwiched between two thin h-BN layers. Their main finding, in one sentence, is : optically silent hBN and hybrid WSe₂-hBN phonons can be observed in Raman scattering when excited in resonance with WSe₂ exciton resonances. The subject has been further studied in the following years [R2,R3,R4] and it has been demonstrated that the polar phonon involved can show some polaritonic characteristics [R4].

We did not anticipate that our work was to be confused or associated with the physics of ECP at the interfaces between two heterostructures [R1-R4]. The following table illustrates some fo the significant and distinctive aspects that set our findings apart from these prior investigations.

	Electron-phonon coupling [R1-R4]	Our current work
The “quasiparticule” probed	An exciton in material 1 and a phonon (or phonon-polariton) in material 2. It requires an heterostructure.	We are probing “pure” hyperbolic phonon-polaritons: guided modes within the layer and surface modes. No heterostructure required.
Excitonic resonances	The Raman signal occurs in resonance with excitonic states [R1-R4]. Single or double resonances appear required.	No excitonic resonances required. This Raman process is much more versatile for probing phPol as it does not require coupling to excitons. That being said, we also demonstrate that enhancements occurs in resonance with the GeSe exciton, as is well-known typical for resonant Raman spectroscopy.
Interfaces	The emergence of a Raman signal is explained by symmetry breaking at the hBN-WSe2 interfaces [R1-R2]. It originates from the interface and is for this reason called “Interfacial Raman scattering” in Ref. [R2]. Indeed, the symmetry of this quasiparticule is very low [R3] compare to that of the phonon or exciton.	The Raman signal originates from both the bulk region and and interfaces of the GaSe. Again, heterostructures are not required. The symmetry of phPols is that of the dielectric environment and is not reduced by coupling to an electronic excitation. We only probe phPols.
Material thickness limitations.	The activation of Raman forbidden phonons occurs in few-layer hBN. They are not activated in thick hBN layers as mentioned in Ref. [R3].	We can study phoPols without restrictions on the thickness of the polariton host material. That being said, backscattering Raman can be most easily observed on layers thinner than about 1 micron. See Supplementary.

Although exciton/phonon-polaritons quasiparticles can be probed using “interfacial Raman scattering” as demonstrated in Ref. [R1-R4], the information extracted appears, up to now at least, relatively limited. **In contrast, we are reporting the full dispersion curve of both hyperbolic guided and surface polaritons in thin GaSe without the need for excitonic resonances or heterostructures, and without thickness limitations.** To avoid confusion, we have added the four references cited above to the manuscripts with a few words of explanation on page 3,

“Recent works have demonstrated that Raman spectroscopy could probe silent polar phonons and phonon-polaritons through the strong cross-material electron-phonon coupling occurring at the symmetry-breaking interfaces of VdW heterostructures under resonant excitation[12-15]. Here, we probe hyperbolic

guided and surface polaritons without the need for excitonic resonances, heterostructures or strong electron-phonon interactions.”

on page 13,

“These resonant Raman results can provide insights into the coupling between excitonic and polaritonic states [15]. Most significantly, the enhancement of the Raman cross-section, even in quasi-resonance conditions, enables a closer study of polaritonic effects that could not be observed away from resonance due to weaker signals. Under stricter resonance conditions, even greater enhancements are expected.”

and on page 14,

“It also includes crystals for which inversion symmetry is broken by a discontinuity or a perturbation such as an external field or an interface [13,15].”

[R1] Jin et al., Nat. Phys, 13, 127 (2017)

[R2] Chow et al., Nano Lett **17**, 1194 (2017).

[R3] Du et al., Phys Rev B **99**, 205410 (2019).

[R4] Viner et al., Phys Rev B **104**, 165404 (2021).

***COMMENT**

Another conclusion of the paper is that the thin nature of the samples allows polaritons to be observed in backscattering not just forward scattering. The breakdown of conservation of forward momentum due to thin samples has been known for a very long time, e.g. it in allowing the backscattering geometry for four wave mixing in GaAs quantum wells.

1- One failing of the paper is that it doesn't sufficiently address the existing literature in the field.

2- This includes the work on Raman scattering from polaritons in exfoliated Van der Waals materials.

RESPONSE

1-Indeed the breakdown of momentum conservation “rule has been known for a very long time” and is ubiquitous in non-linear optics. Since Raman spectroscopy can be interpreted as a third order non-linear effect, it applies as well. The theory of Raman scattering from thin layers (Eq.1) was published more than 4 decades ago (starting from 1976 with many contributions in the following decade). We believe that the references provided in Supplementary and those in the main text ([22] and [23]) are more than sufficient.

2- See previous and following responses.

***COMMENT**

In addition the paper doesn't review the literature on GaSe Raman scattering which includes experiments on 'thin' layers, e.g. Selective Raman modes and strong photoluminescence of gallium selenide flakes on sp² carbon Journal of Vacuum Science & Technology B 32, 04E106 (2014); <https://doi.org/10.1116/1.4881995>. Including a review of this work in the paper would enable the reader to have a better sense of the novelty within the results.

RESPONSE

In selecting the 35 or so references for this manuscript, we certainly aimed at citing the most relevant literature relating to phonon-polaritons, Raman, and GaSe.

Ref. 10.1116/1.4881995 indicated by the reviewer reports the activation of E'(LO) phonon at the edges of GaSe crystals. This phenomenon is explained by the breakdown of selection rules at crystal edges. This phenomenon can be observed in most 2D flakes. As suggested by the reviewer, we have added this reference on page 4 and added to the text that signals did not share the same origin.

We have also added 3 recent references on GaSe.

COMMENT

In conclusion, the paper presents some nice experiments which are well presented but these results are not sufficiently novel for publication in Nature Communications. The existence of phonon polaritons in GaSe is already established; the fact that Raman scattering can be measured from these modes is established; the fact that the dispersion of such modes can be measured by angle dependent Raman scattering is established. I therefore advise the editor not to accept this article for publication and the authors to revise the manuscript to include all the relevant related literature and publish this work in a reputable archive journal.

RESPONSE

Here, reviewer #3 forcefully reiterates his core message, but without adding further evidence. As we have established above, our work is very distinct from those cited by the reviewer, both in terms of the physics involved and, most importantly, the impact it may have of the development of phonon-polaritons devices.

Reviewer #4 (Remarks to the Author):

***COMMENT**

In this manuscript, the authors report probing phonon polariton in GaSe thick flakes and want to demonstrate that routine Raman spectroscopy is sufficient for acquiring information for phonon polaritons in certain materials. This manuscript is a well-prepared Raman study on GaSe phonon polariton, including experimental measurement and theoretical calculations. Especially in the supporting information part, the authors provide rich details about the calculation, which is a good reference for those dealing with Raman spectral calculations.

RESPONSE

We would like to thank Reviewer #3 for evaluating our manuscript and preparing this detailed report. We would also like to apologize for taking several months for providing this response. The first author is a former Ph.D. student that graduated more than one year ago. Despite his strong commitment and desire to see the core of his PhD findings published, it has not been an easy task to collaborate on revising and improving the manuscript with a regular full-time job in a start-up and a young family.

***COMMENT**

However, the experimental evidence in the current manuscript does not fully support the conclusion.

RESPONSE

To better convince the reviewer, we have added the dispersion curve measured on five samples in Fig. 1(b) (please note that the Figure numbering has changed). Although not all surface and guided modes are observed from all samples, the surface and guided polaritons are clearly identifiable. In addition, this experimental data demonstrate that the dispersion relation is strongly influenced by the sample thickness and evolved as expected for polaritons. In our opinion, the data presented in this work strongly support the main message and conclusion: backscattering Raman spectroscopy probes the dispersion curve of hyperbolic phonon-polaritons in Van der Waals crystals.

The following text was added.

“Similar Raman spectra as a function of sample tilt were acquired in the manner presented in Fig. 1a) on samples of different thicknesses, from 201 nm to 12 microns. Observed spectral line positions are reported in Fig. 1(b) as a function of the in-plane propagation wavevector (k_{\parallel}). These dispersion curves are organized in three groups (blue, green, yellow) according to their frequency range. Sample thickness has a particularly strong influence on dispersion of the low and high energy groups, a behavior again incompatible with purely mechanical phonons. The data of Fig. 1(a-b) originate from the backscattering of light from phonon-polaritons (phPols). The strong polaritonic character is further corroborated below through 1) Raman spectra simulations, 2) experimental data

demonstrating strong phPol confinement, and 3) experimental resonant Raman data demonstrating enhanced PhPol scattering cross section for excitation close to the GaSe $1s$ exciton.

***COMMENT**

The authors present the measured Raman spectra of GaSe as a function of impinging angle in the beginning, which shows the features in the Raman spectra changes with k indeed. However, the rest of the paper, or the essentials of the discussions, are based on pure calculations. The only direct comparison of measurement and calculation in Fig.2e does not show a good agreement, although the authors argued in the main text for the possible causes.

RESPONSE

We are surprised by this comment, since all 4 figures of the original manuscript included experimental data: Fig. 1(d), 2(e) and 4 presented more than 14 raw Raman spectra, Fig. 3 compiled the experimental data collected on more than 15 samples, and Fig. 4 experimentally demonstrated the capacity of resonant excitonic excitation to enhance the Raman scattering efficiency. We are an experimental research group and this is an experimental paper. That being said, we strive to analyze our results using calculations. These calculations are there only to help understand and interpret the data. We do believe that this paper would remain novel and relevant to the Nat. Comm. readership without the Raman simulations. However, we believe that the two together make for a deeper and stronger study.

To avoid any confusion on this point, we have reorganized the figures to make sure that the experimental data stands out much more. As mentioned above, we have added experimentally measured dispersion curves measured on 5 additional samples (Fig 1(b)). Three panels relating to calculations were moved to the Supplementary. In the current version, the calculated spectra in Fig. 2(a-b-c) and Fig. 3(a) are used to support for the interpretation of the experimental data.

***COMMENT**

This is a good paper showing an interesting case for capturing a few polariton features in two-dimensional materials using a conventional spectroscopic tool, Raman spectroscopy. Nevertheless, the connection between the measurement and calculation is not sound enough to be extrapolated to study phonon polariton properties offered by SNOM extensively.

RESPONSE

In our presentation of our approach based on Raman spectroscopy, we certainly did not want to minimize the relevance or contributions of s-SNOM techniques, from which the science of polaritons in 2D materials has strongly benefited. Applied to phonon polaritons, s-SNOM has been perfected for more than a decade. As we tried to emphasize in our introduction, Raman is a complementary tool and together, may help the development of the science and applications of phonon-polaritons. From a classical

perspective, two important mechanisms relate to light-matter interactions: absorption/emission and scattering. These two mechanisms provide a different but valuable perspectives that has been both exploited in too many fields to count. Hence, Raman complements the absorption/emission experiments based on s-SNOM.

Although our Raman data may not match in the eye of the reviewer the data provided by s-SNOM, Raman spectroscopy captures more than just a few features: we do report full dispersion curves (Fig 1(a-b)), report several guided and surface modes (Fig 2(c) and Fig 3(B)), demonstrate the effect of confinement (Fig. 3(a)), and use resonant with the exciton to enhance the signal.

In this work, Raman calculations are used as a guide to interpret the data. The fact that the calculation does not provide a quantitative match to the experimental data does not decrease the value of the experimental data. Nonetheless, the calculation is, at this stage, a helpful and valuable tool for the discussing polariton scattering.

Modifications to the text have been made to emphasize the fact that the Raman simulation is mainly used as a qualitative tool for the interpretation of the experimental data. The discussion on the nature of the discrepancies in Fig. 2(e-f) (previous version) is now better described. In reference to the two panels of Fig 2(c), the following text has been added.

“At 45°, four UEp contribute to the calculated spectra (dotted blue curve). These modes have a dominant L_e coordinate (see Fig. \checkmeSeb{SF4}) and the two high- k_{\parallel} UEp dominate the spectra with little contribution from the LSp. Experimentally, a weak LSp can be identified and the two dominant UEp appear as a single line (black curve), due in part to a limited signal-to-noise at high measurement angles. Because of the low dispersion at high θ (high k_{\parallel}), the correction for the finite acceptance angle does not significantly affect the spectra (red curve).} At $\theta=5^\circ$, the calculated Raman spectrum (blue curve) does not match the experimental spectra, but the calculated polariton spectra corrected for the acceptance angle (red curve) considerably improve the agreement with the experimental spectrum (black curve). It is important to note that the calculated spectra do not depend on any free adjustable parameters (see Section S9) and the agreement with measured spectra is only qualitative at low k_{\parallel} , since the linewidths of the LSp and UEp branches are respectively underestimated and overestimated. More work is required to improve the predictive power of Raman intensity calculations at low incidence angles, which is beyond the scope of the present work. In the following, Raman calculations are used as a qualitative tool for the identification of pPhPs and analysis of their spectra.

***COMMENT**

The supporting information pointed out, "For typical phonon frequencies ($\omega \approx 200\text{--}1000$ cm⁻¹, polaritonic effects are observed for wavevectors values of the order of $k \sim 1 \times 10^3$

cm⁻¹." As known, the wavevector for the phonon polariton of a 220 nm thick MoO₃ is kpolariton $\sim 1 \times 10^5$ cm⁻¹, two orders larger than the authors' description. (Nature Communications 11, 2646 (2020)). The authors are encouraged to provide dispersions of the phonon polaritons by calculating the imaginary part of the Fresnel reflection coefficient of the Air/GaSe/SiO₂/Si multilayer system (Nature Nanotech 12, 207–211 (2017)). If the kpolariton is close to or not much smaller than the wavevector of the light, the propagation directions of q_s and q_i in Fig S1c will deviate greatly, and there will be a significant difference between the reflection angles (θ_{os}) and the incident angles (θ_{oi}). When obtaining the phonon's momentum, authors can plot the θ_{os} at different θ_{oi} in Fig S1c.

RESPONSE

The discussion in supplementary initially referred to bulk polaritons, where the light line indeed crosses the phonon frequency around 10³ cm⁻¹. To avoid confusion, we have added a reference to an early work on phonon-polaritons and a paragraph that explain why k-values in thin van der Waals samples reaches much higher values (10⁵ cm⁻¹).

Indeed, wavevectors involved in our studies are of the order of 10⁵ cm⁻¹. As suggested by the reviewer, we have,

- 1) added the reflection data that demonstrates that the Raman simulation adequately reports the dispersion curves. See Fig. 2(a). In fact, the agreement between Raman and reflectivity data is perfect, except that the Raman intensity is a much more powerful for the analysis of our experiment data.
- 2) converted the tilt angle to wavenumber. As can be seen, the range of wavenumbers (the in-plane projection), varies from 0 to 10⁵ cm⁻¹. See Fig. 1(b) and 2(a-c).

The following text was added,

“Resonances in the imaginary part of the reflectivity coefficients are indicated by the dashed blue lines in Panel (a). This more conventional way to calculate surface and guided polariton resonances perfectly agrees with the Raman calculation, but the latter has the advantage to report on the selection rules and the relative intensities relating to scattering.”

*COMMENT

A detailed description of the experimental configuration is missing, including how to adjust a good focus at large oblique angles and how the 15-degree impinging angle is estimated. When the incident angle θ in Fig 1a is 45°, one long-distance objective alone may not collect sufficient Raman scattering signal from the reflection light.

RESPONSE

Reviewer #4 is right, angle-resolved spectroscopy can be challenging: the excitation spot becomes ellipsoidal, fine focal adjustments may be required, the collection efficiency decreases and, for some samples, it may be difficult to reach the highest mechanically allowed angle due to a reduced signal-to-noise. For these measurements,

we have fabricated a special vacuum sample cell and measurements up to 45 degrees can routinely be achieved. It is relatively easy to measure Si, GaAs and GaSe phonons at 45 degrees and, through an optimized experimental alignment protocol, we can observe polaritons! Although not routine, high angle measurements are not unusual either in optical spectroscopy. We have added the following sentences to the METHODS section:

“The sample rotation axis is perpendicular to the incident light beam and positioned at the laser focal point on the sample surface (see inset of Fig. 1(b)). A goniometer reports the tilt angle with an uncertainty of at most 0.5 degree over the whole angular range.”

***COMMENT**

The double peak feature for UEp1 and UEp2 is unclear in the experimental data in Fig 1d and Fig 2e, while the calculation predicted a clear double peak feature. In Fig 2f, the authors pointed out, "data points represent the dominant frequencies identified using a Lorentz-line shape analysis of the experimental spectra." In contrast, the calculated spectra did not show Lorentz-lineshape (245 cm⁻¹ to 255 cm⁻¹) with the sample tilting angle from 0° to 10°. The author may consider taking the second derivative of the spectrum to locate the peak positions of UEp1 and UEp2.

RESPONSE

As suggested by the reviewer, we have attempted several numerical strategies to bring out the underlying substructure, including a second derivative. Unfortunately, this did not prove advantageous: the derivatives “broaden” the signal and amplify the noise presents in the experimental data (despite using noise-reducing strategies).

Higher order guided modes have been so far seldom observed, either due to the instrumental broadening or sample quality. That being said, we present in Fig 3(a) and 3(b) data from a relatively thick sample where two extraordinary guided polaritons could unambiguously be resolved (see dotted box in Fig.3(a)). The four experimental spectra shown in Fig. 3(b) present the behavior of these two modes at two different temperatures and two excitation wavelength. The text was modified to bring out this aspect more clearly.

Future work will include improved sample preparation methods, increased angular resolution, and improved instrumental sensitivity. This may allow the observing higher order modes more consistently.

***COMMENT**

Although Raman spectroscopy shows advantages in detecting phonon polaritons from the far-field, the precision of the dispersion curve depicted by Raman spectroscopy is far from the level s-SNOM can offer at the moment. In conclusion, the authors should discuss possible improvement routes for future technologies based on the above shortcoming of Raman spectroscopy.

RESPONSE

Indeed, the Raman spectroscopy of hyperbolic polaritons in Van der Waals crystals is relatively new and does not match the performance of today's s-SNOM experiments. As suggested, we have added a few comments on possible improvements in the conclusion.

“The limited wavevector resolution can be improved by reducing the angular acceptance of the incoming and outgoing light, as demonstrated in S14, or implementing k-imaging techniques for simultaneously resolving both the frequency and propagation wavevector. “

The conclusion also discusses how resonant excitation can be used to improve the scattering efficiency. In our opinion, Raman spectroscopy of phonon polaritons can undoubtedly be improved.

*COMMENT

In Fig4, the authors should briefly discuss the physical origin of temperature-induced Raman shifts and cite relevant new literature, such as Sci Rep 6, 32236 (2016). Since several textbooks elaborated well on the observation of probing polaritons in GaAs at low temperatures, what is new in Fig.4 should be stressed.

RESPONSE

The temperature-induced phonon shifts have been measured and modeled in Mat. Res. Bull. 37, 169 (2002). Since this work predates the reference suggested by Reviewer 4 and directly relates to GaSe, it is the reference that was added to the revised version of the manuscript. In addition, the following sentence has been added.

“As demonstrated in Ref. [35], temperature-induced phonon shifts result from thermal expansion and three-phonon processes. In the following, however, we analyze the effects of temperature on the Raman intensity, which result from resonant excitation conditions created by temperature tuning the 1s-exciton energy with respect to the laser energy.”

Fig. 4 is very important in the context of our study on phonon-polaritons:

- 1) It demonstrates that two guided mode polaritons, UEp1 and UEp2, can be clearly resolved.
- 2) Through temperature-tuning, the GaSe exciton is brought into quasi-resonance with the laser excitation (633 nm).
- 3) It demonstrates a singular enhancement of the UEp1 and UEp2 polaritons. This is a means to compensate for the low-efficiency traditionally associated with Raman spectroscopy.
- 4) It reveals a third-guided polariton, LOp, which could not be observed without quasi-resonant excitation. This polariton is not observed with above gap radiation (532 nm).

- 5) Resonant Raman spectroscopy is a powerful technique that will be used in future work to study the coupling between phonon-polaritons and excitons.

These results are novel and further demonstrated the interest of Raman spectroscopy of confined hyperbolic polaritons in Van der Waals crystal. The ending paragraph of this section has been modified. It now reads,

“These resonant Raman results can provide insights into the coupling between excitonic and polaritonic states. Most significantly, the enhancement of the Raman cross-section, even in quasi-resonance conditions, enables a closer study of polaritonic effects that could not be observed away from resonance due to weaker signals. Under stricter resonance conditions, even greater enhancements are expected.”

***COMMENTS**

- In Fig 2f, the green dashed line is missing but mentioned in the caption.
- Quote from the SI, "phonons can straightforwardly be defined by their A, E, LO and LO characters." LO duplication.
- Quote from the sentence, " The calculated polariton scattering intensity is shown in 2(a-c,d,e) as a function of sample tilt angle" should be 2(a-c,d,f)
- Several typos, such as SiO₂ printed as Si0₂.
- On page 14, the first sentence is incomplete.

RESPONSE

We are thankful to the reviewer for pointing out these mistakes and irregularities. They have all been corrected. Please note that panels (a-c) were moved to supplementary.

REVIEWER COMMENTS

Reviewer #1 (Remarks to the Author):

The revised manuscript by Bergeron et al reports on Raman spectroscopy of volume-confined hyperbolic phonon polaritons in broken-inversion GaSe thin films. The authors addressed all of my comments very comprehensively, and the respective revisions have increased the clarity of the paper. In particular Fig. 1b is very convincing now as to the major claim of the paper.

The one point I am a bit hesitant about, still, is the claim of volume-confined modes being observed at 12 μm thick samples. This is a very bold claim which in fact I think is unnecessary for the given context. Following the trends in Fig. S7, the dispersive character washes out for very thick samples, and the "last" remaining resonance converges to the LO phonon resonance at 255 cm^{-1} . One could argue that this means that the character of the mode at this thickness is non-dispersive, i.e., really loses its "light-like" propagating polaritonic character and essentially is a LO phonon. In the light of this picture, the claim of volume-confined modes being observed at such thicknesses is very questionable. The dispersion of this mode in Fig. 1b is negligible (I would think), consistent with this picture. In other words, with vanishing dispersion, the experiment cannot differentiate between a polariton and a phonon, and thus cannot proof the volume-confinement.

If the above issue is resolved, I now support publication of their manuscript in Nature Communications.

Minor points:

- The horizontal axis label in Fig S5 is missing.
- it would be nice to see the direct comparison of experimental results in Fig. 1b and the respective calculations. This doesn't have to shown their, SI would also suffice. But in principle, "underlying" the respective calculation would be even more convincing.

Best Regards,
Alex Paarmann

Reviewer #2 (Remarks to the Author):

The authors have responded to the raised concerns in a thoughtful, and meaningful way. I am sufficiently convinced that their results support their arguments, and believe the work is sufficiently novel. I therefore endorse its publication in Nature Communications.

Reviewer #4 (Remarks to the Author):

The authors have made significant revisions and have mainly addressed my concerns in the review report. As I commented in the last report, this paper showed a Raman spectroscopy approach to investigating phonon polaritons in GaSe. Scientifically and technically, the findings and the methodology are inspiring.

The current popular materials for phonon polariton investigations are hBN and MoO₃, which exhibit different hyperbolic dispersions. Why did the author choose to study GaSe? Can the methodology also be applied to studying hBN and MoO₃? If the authors can do that, there are abundant references with other research techniques to compare directly, and the work will attract more attention.

Reviewer #1 (Remarks to the Author):

COMMENT

The revised manuscript by Bergeron et al reports on Raman spectroscopy of volume-confined hyperbolic phonon polaritons in broken-inversion GaSe thin films. The authors addressed all of my comments very comprehensively, and the respective revisions have increased the clarity of the paper. In particular Fig. 1b is very convincing now as to the major claim of the paper.

RESPONSE

We were very pleased to know that Reviewer #1 finds the data “very convincing”.

COMMENT

The one point I am a bit hesitant about, still, is the claim of volume-confined modes being observed at 12 um thick samples. This is a very bold claim which in fact I think is unnecessary for the given context.

RESPONSE

We agree with Reviewer #1 on both aspects: 1) For the 12 um sample, the phonon-component of the polariton likely dominates the photon-component, and 2) this particular data is not required to support the major claims of the paper.

We have chosen to leave the data in Fig. 1, but have made two important modifications to the manuscript. The following two statements,

1-“Similar Raman spectra as a function of sample tilt were acquired in the manner presented in Fig. 1(a) on samples of different thicknesses, from 201 nm to 12 um.”

2- “GaSe is a promising material for polaritonic applications. First, confined phPols are observed in rather thick samples, up to 12 um, suggesting relatively low propagation losses.”

now read as,

1-“**Similar Raman spectra as a function of sample tilt were acquired in the manner presented in Fig. 1a) on samples of different thicknesses, from thin samples (201 nm) up to very thick samples (12 um) where volume-confined polaritons are not expected.**”

2- “**GaSe is a promising material for polaritonic applications. First, confined phPols are observed in rather thick samples, up to at least 750 nm, suggesting relatively low propagation losses.**”

In addition, we have removed the “12 um” mentioned in the abstract and replaced it by the “750 nm”.

COMMENT

If the above issue is resolved, I now support publication of their manuscript in Nature Communications.

RESPONSE

We believe that the modifications presented above resolve this issue. We would like to thank to reviewers #1 for having helped us clarify this important aspect.

COMMENT

Minor points:

- The horizontal axis label in Fig S5 is missing.
- it would be nice to see the direct comparison of experimental results in Fig. 1b and the respective calculations. This doesn't have to shown their, SI would also suffice. But in principle, "underlaying" the respective calculation would be even more convincing.

RESPONSE

- The horizontal axis label has been added. Thanks for noticing this.
- Indeed, it would have been very nice! However, modeling the effects of angular broadening at low wavevectors is not yet adequately perfected and would remain largely qualitative. This is the subject of ongoing work, both on the theoretical (better treatment of angular broadening) and experimental (better angular resolution) side.

Reviewer #2 (Remarks to the Author):

COMMENT

The authors have responded to the raised concerns in a thoughtful, and meaningful way. I am sufficiently convinced that their results support their arguments, and believe the work is sufficiently novel. I therefore endorse its publication in Nature Communications.

RESPONSE

Thanks to reviewers #2 for having participated in the review process and helped us improve the manuscript.

Reviewer #4 (Remarks to the Author):

COMMENT

The authors have made significant revisions and have mainly addressed my concerns in the review report. As I commented in the last report, this paper showed a Raman spectroscopy approach to investigating phonon polaritons in GaSe. Scientifically and technically, the findings and the methodology are inspiring.

RESPONSE

A special thanks for Reviewer #4 for this these kind words.

COMMENT

The current popular materials for phonon polariton investigations are hBN and MoO3, which exhibit different hyperbolic dispersions. Why did the author choose to study GaSe? Can the methodology also be applied to studying hBN and MoO3? If the authors can do that, there are abundant references with other research techniques to compare directly, and the work will attract more attention.

RESPONSE

We have modified the following sentences to further bring out the interest in GaSe:

Page 3 : **“For this demonstration epsilon-GaSe is selected due to its strong polar resonances and non-linear characteristics, nested Reststrahlen bands, double type-II hyperbolic regions (Fig. 1(a))”**

Page 3: **“These nonlinear properties contribute to the Raman scattering efficiency from long-range polarization waves, making GaSe an ideal prototypical system for studying phonon-polaritons.**

Naturally, it is beyond the scope of this work to study other materials of high interest, including h-BN and MoO₃, as such study can take more than a year. That being said, the latest version of the manuscript includes 12 references directly citing papers on these two materials (Refs 2, 4-11, 19, 37-38), thereby directly connecting our work to this important body of literature.

Finally, we have added these sentences to the conclusion that further establish the relevance of our work to other 2D materials.

“It can be applied to a large family of Van der Waals crystals [39] with few restrictions on polariton wavevector (> 15000 cm⁻¹) and frequency (from 10 to more than 2000 cm⁻¹).”

and

“It also includes crystals for which inversion symmetry is broken [40] by a discontinuity, a perturbation such as an external field, or an interface [13,15].

New references were added to support these aspects (Refs 39 and 40).